# Plastic Creep Constraint in Nylon Instrument Strings

**DOI:** 10.3390/ma18020223

**Published:** 2025-01-07

**Authors:** Nicolas Lynch-Aird, Jim Woodhouse, Claire Y. Barlow

**Affiliations:** 1The Old Forge, Burnt House Lane, Battisford, Stowmarket, Suffolk IP14 2ND, UK; 2Department of Engineering, University of Cambridge, Trumpington St, Cambridge CB2 1PZ, UK; jw12@cam.ac.uk (J.W.); cyb1@cam.ac.uk (C.Y.B.)

**Keywords:** nylon, string, instrument string, musical string, viscoelastic, creep, plastic creep, creep constraint, crystallinity

## Abstract

A number of rectified nylon harp strings, having the same nominal diameter, were subjected to different sequences of applied stress steps. Each string was tested continuously for several weeks to allow sufficient time for the stretching responses to be clearly observed. Qualitatively, much of the observed behaviour was in accordance with established expectations. However, the quantitative data gathered here are believed to be novel, and revealed some surprises. The strings displayed a combination of elastic stretching, fully recoverable viscoelastic stretching, and apparently non-recoverable plastic stretching. The elastic and recoverable viscoelastic stretching behaviour was quite straightforward, but the plastic creep behaviour was more complicated, with a number of the strings displaying an unanticipated phenomenon. When the applied stress was left unchanged, or was stepped down and back up again, it was noticed that, in some cases, the extent of the subsequent plastic stretching, when the applied stress was next increased beyond its previous maximum, was significantly less than might have been expected. The tests revealed that this apparent plastic creep ‘constraint’ mechanism seemed to depend primarily on the length of time between successive overall rises in the applied stress, with a threshold somewhere in the range of 30–40 days. It is suggested that this phenomenon may be due to a gradual increase in the polymer crystallinity during this rest period. Two of the strings, which were tested over a wider range of applied stress levels, revealed another aspect of the creep behaviour. There appeared to be an initial ‘straightening’ phase during which the plastic stretching rose with the applied stress in a diminishing manner to reach a stretching limit. As the applied stress was increased, this initial straightening was overtaken by an unlimited main stretching phase, which rose slowly at first before approaching a linear increase with the applied stress.

## 1. Introduction

When a nylon musical string is subjected to a jump in the applied strain, it exhibits an immediate elastic response, followed by a slower viscoelastic relaxation. A musician would describe the string as being tuned up, and then gradually slipping a little flat, or conversely being tuned down, and then gradually recovering to become a little sharp. Alternatively, the string could be subjected to a jump in stress. This would correspond to the musician tuning to a higher pitch, then making further adjustments as necessary in order to keep the pitch constant. Those further adjustments, and their pattern over time, constitute an example of material creep [1].

This article presents experimental results showing some details of this viscoelastic behaviour when strings are put through various sequences of increasing and decreasing stress. The observed behaviour is in qualitative agreement with intuitive expectations, but some intriguing details will be revealed. A number of rectified (centreless ground) nylon pedal harp strings were subjected to different sequences of tensile stress steps, with each step typically lasting about a week. So far as the authors are aware, no comparable quantitative measurements have previously been published. A survey of the literature found no similar tests of instrument strings. Equivalent tests of badminton and tennis racket strings have usually been run for no more than a few days and typically have taken the form of a single-step creep or relaxation test [2,3,4,5].

The stepped-stress tests of this present study arose out of a larger series of studies investigating the mechanical properties of synthetic polymer and natural gut harp strings [6,7,8,9]. Those studies required each of the strings to be tested at multiple target frequencies, and to have settled, as far as could be determined, prior to each set of tests. It was quickly found that each time the tensile stress, or the temperature, was increased beyond the previous maximum, the string would start creeping again, and that it would typically take weeks to settle.

The initial stepped-stress studies to be described here were motivated by a need to better understand the string creep behaviour, and also a need to investigate how the tensile Young’s modulus of the strings varied with the applied stress [6]. Over time, techniques were developed both to accelerate the string settling time, and to reliably determine when the string had, at least for the purposes of the measurements being made, stopped creeping [6,8]. The results from the first three stepped-stress tests were also provided to successive undergraduate students in the Engineering Department at Cambridge University (U.K.) for use in their final-year projects [10,11,12]. Their aim was to develop a model that might explain the measured creep behaviour.

It became apparent that the measured responses exhibited two quite different modes of creep behaviour [11]. When the applied stress was varied up and down, but without exceeding the historic maximum, the strings appeared to display the sort of recoverable creep often observed with viscoelastic materials [13,14]. These creep episodes showed a high degree of similarity in response to positive and negative stress steps, both in terms of the response amplitudes and their time constants.

When the applied stress, or temperature, exceeded the historic maximum by more than some threshold level, however, the strings appeared to undergo a different and more extensive form of stretching, which appeared to be non-recoverable. This will be referred to here as ‘plastic’ creep, though it should be noted that it was not clear if the associated deformation was truly permanent, or if it might in fact have been very slowly recoverable.

Eventually, it was noticed that the extent of the plastic creep appeared to vary depending on whether the string was already creeping when the applied stress was further increased. If the string was already undergoing plastic creep, then the subsequent episode of plastic creep was of a greater magnitude than if the string had been allowed sufficient time to settle.

This paper presents the results from the full set of stepped-stress tests. The observed behaviour is described, highlighting the magnitude variations in the plastic creep episodes, and how these appeared to change with the test parameters. A succession of constitutive models are then used to further explore the different aspects of the measured strain responses.

## 2. Underlying Physics of Polymer Behaviour

Before showing the detailed results, it is helpful to summarise the accepted understanding of how an extruded nylon string should behave in response to step changes in stress or strain. A nylon string consists of a tangle of long molecules, with a significant degree of alignment from the extrusion process. Stretching the string by placing it under tension will increase this degree of alignment. As a result, the longitudinal Young’s modulus of the string will be greater than that of amorphous bulk nylon, and it will increase further as the degree of molecular alignment increases [15]. The molecular backbone is formed of strong covalent bonds, which remain intact throughout, and the Young’s modulus is governed by the force required to stretch and rotate these covalent bonds as the molecules are straightened by a small increase in tensile stress. If a musical string is tightened towards its limiting tensile strength, the increase in Young’s modulus is very significant, as has been documented in the previous studies [6,7].

There are relatively weak hydrogen bonds and van der Waals bonds between adjacent molecules, tying them together at certain points and stopping them from sliding past each other; without these, the bulk material would flow like a liquid in response to tensile stress. Suppose there is a step change in tensile strain, when the string is stretched by an additional amount and the length is then kept fixed. In the immediate aftermath of the step there will be a distribution of residual stress within the body of the string, with some molecules under more tension than others. As a result, the hydrogen and van der Waals bonds are under varying degrees of stress, with some statistical distribution.

From time to time, random thermal perturbations will nudge one of these bonds beyond its limiting strength. It will break, allowing the pattern of stress in the surrounding molecules and bonds to relax slightly. This thermally activated relaxation of internal stress is the mechanism of the slow creep recovery after the strain step. If the temperature remains constant, there is a limiting level that the magnitude of these thermal perturbations very rarely exceeds. As a result, after some time, all bonds that are within reach of being broken will have done so. This is the reason the creep process eventually slows down and stops. However, if the temperature is then increased, the thermal perturbations increase in magnitude, and further creep will occur.

Now suppose the tensile strain is stepped downwards, then after a while it is stepped back up to the previous level. The downward step will tend to stress many of the hydrogen and van der Waals bonds in the opposite direction to the upward step, and some of these will gradually relax out by the thermal process just described. When the strain steps back up to the previous level, one intuitively imagines that some of those bonds resume more or less the same state of stress as before—but this is already known to be below the threshold for thermal activation. So, one would expect that under these circumstances, once the relaxation process is complete, the string will return to very much the same state as before. But if the upward strain step exceeds the previous maximum, many more hydrogen and van der Waals bonds would be brought within range of the thermal perturbations, and additional creep can be expected. A very similar argument can be applied to the situation in which it is the stress rather than the strain which is varied in a stepwise manner.

## 3. Methods and Materials

The two test rigs used for this study, one of which is shown in Figure 1, were the same as those used for the studies investigating the mechanical properties of harp strings, with a detailed description being given in [6]. A single string was mounted about 2 cm above a solid hardwood baseboard, with a load cell at one end, to measure the string tension, and a motorised string winder at the other. The motor driving the string winding shaft incorporated a shaft-mounted optical rotation sensor. The output of this, combined with knowledge of the worm gear reduction ratios and the diameter of the winding shaft, enabled the lengths of string wound on and off the winding shaft, referred to as the string tuning length adjustments, to be determined with an accuracy of better than ±0.008 mm.

Both the sensitivity and the zero offset of the load cell (Novatech F256EFR0KN 40 kg) varied with temperature, as did the gain of the external electronics. Digital thermometers were used to monitor the temperature of the load cell and inside the electronics module and, after thermal characterisation, the load cell could be used to measure string tensions over a range of 0–300 N, with an accuracy of ±0.2 N.

The string passed over two bridge pins set 50 cm apart: one mounted horizontally, close to the winding shaft, and one mounted vertically at the load cell end. The bridge pins were mounted perpendicular to each other in order to minimise the variation in the effective string vibrating length according to the direction in which the string was plucked [6,16]. The string was plucked at its mid-point, and a microphone was used to record the pluck responses.

The temperature inside the test chamber was monitored using a digital temperature and humidity sensor attached to the plucker assembly. The original sensors (Sensirion SHT75) had a nominal accuracy of ±0.3 °C and ±1.8% RH (relative humidity). Eventually, one of these failed and the sensors in both test rigs were replaced with updated versions (Sensirion SHT85) with an improved nominal accuracy of ±0.1 °C and ±1.5% RH. The temperature was controlled by switching on different combinations of incandescent light bulbs. Black plastic drinking straws mounted below each light bulb acted as crude light guides leading to photo-transistors for monitoring the light bulbs. In this way, the control program could attempt to switch in an alternative light bulb or bulb combination when a bulb failed. Fans and a heat distribution bar, positioned just below the string, helped to maintain a constant temperature throughout the test chamber and along the length of the string.

The cut-off water bottle at the back of the chamber provided a crude form of humidity control, maintaining the relative humidity at a level of about 60%, when the temperature inside the chamber was 20 °C. A water level monitor coupled to an external reservoir (a car windscreen washer-fluid bottle) enabled the water level to be maintained automatically to within 1 or 2 cm. The shallow plastic tray shown in the back left corner of the test chamber could be used to temporarily boost the humidity levels (adding water to the tray increased the surface area for evaporation [8,9]) but was not required for this study.

### 3.1. Strings Tested

A total of eight rectified nylon harp strings from Bowbrand (Norfolk, U.K. [17]) were tested, each having a nominal diameter of 0.66 mm. The diameters of the strings were measured prior to testing using a manual micrometer screw gauge with a resolution of 0.01 mm. Multiple measurements were taken along each string and averaged. The string lengths were measured using a tape measure with a resolution of 0.5 mm. The strings were also weighed in the Engineering Department at Cambridge University. The bulk densities of the strings were then obtained from these mass, length, and diameter measurements.

As described above, the test rigs could measure the string length adjustments and tension directly, while the string frequency could be determined from the Fourier transform of the recorded pluck responses. Once each string had been mounted in its test rig, the string diameter could only be remeasured infrequently, using the same manual micrometer as for the original diameter measurements, and only by taking the perspex front off the test chamber. It was therefore not possible to continuously monitor the tensile stress applied to the string.

For the first test, which was run very early in the wider string testing programme, a choice therefore had to be made between maintaining the string at constant tension or constant frequency for the duration of each ‘stress’ step. The stress σ could be calculated either as the ratio of the tension *F* to the string cross-sectional area *A*, or from the string fundamental frequency $f_{1}$ using the well-established equation for the frequency of an ideal flexible string [18]: (1)$$\sigma =F/A=4f_{1}^{2}L_{V}^{2}\rho$$
where ρ is the bulk density and $L_{V}$ is the vibrating length of the string. On the basis that the string density should vary less than its diameter as it was stretched, it was decided that the closest approximation to applying a constant stress would be to run the test rig in its constant-frequency mode, changing the target fundamental frequency for each stress step. This approach was subsequently validated when measurements showed that, following an initial settling period, for strains above about 6%, the bulk density of nylon strings remains constant as the stress or strain is increased, at about 98% of the unstretched density [6]. As will be seen, the majority of the tests applied strains of well above 6%, so maintaining a constant frequency was a good analogue for applying a constant stress.

Different sequences of applied tensile stress steps were used for each test. Table 1 lists the tested strings and gives their unstretched diameters and bulk densities, together with the test rig used, the sequence of target test frequencies used to apply the stress steps, and the dates when the tests were run. The tests fall into two groups: the first three tests with strings N7, N20, and N21 were run over the winter periods between 2012 and 2015, and supplied to the Cambridge University Engineering students for their final-year projects [10,11,12]. A second group of tests were run in 2023 and 2024 to further investigate the observed behaviour. The test with string N27 was focussed on exploring the behaviour when the applied stress did not exceed the previous maximum, while the tests with the other four strings investigated the plastic creep that occurred when the applied stress did exceed its previous maximum, and how the extent of this plastic creep varied with other details of the applied stress sequence and timing.

### 3.2. Test Protocols

The tests were run at a constant target temperature of 20 °C. During most of the tests, combinations of 100 W and 150 W incandescent light bulbs were used to provide heating. The minimum heating adjustment step was thus usually 50 W, which corresponded to a temperature change within the test chambers of a little over 3 °C. The acceptable temperature range was therefore set at 20 ± 2 °C. The test rig control program included a facility to monitor and automatically adjust the heating level to maintain the temperature within a specified range, but this capability was not added until late in 2013, after the first test with string N7 had been run. For this first test, the temperature had to be monitored and controlled manually, which led to the string being subjected to a slightly wider range of temperatures.

At step 10 during the third test (string N21), an additional burst of heating was applied to see what effect this would have. The temperature was increased to between around 40 °C and 50 °C for 3.5 days, while the applied stress was kept at the same level as for the previous step.

To apply the required sequence of stress steps, at the start of each step the string was tuned to the specified target fundamental frequency, and then regularly retuned to that frequency while the string tension and the tuning length adjustments were recorded. For the first test (string N7), the retuning and recording interval was set to one hour, but for all the other tests except string N20 (see below), it was reduced to five minutes. At the end of each step, the test chamber was opened and the string diameter was remeasured.

Typically, each stress step was applied for about a week, which was sufficient time for any relatively fast strain variations to settle down, usually leaving the strain increasing or decreasing slowly at a fairly steady rate. For all strings except N7 and N28, the first stress step was much larger than the subsequent steps and that step was run for a longer period of 11 to 14 days. An exception to this was string N34, for which the first step was applied for six weeks, to match the duration of the first five stress steps applied to string N21.

During the first three tests, measurements were also made of the longitudinal Young’s modulus. For strings N7 and N21, the Young’s modulus was measured at the end of each step by varying the string tension up and down, relative to the starting tension, in the form of a triangle wave [6]. For string N7, the tension was varied over a range of ±10% in 2% steps, while for string N21, the tension modulation range was reduced to ±6% in 1.2% steps. For string N20, the Young’s modulus was measured for the whole period of each step by modulating the target frequency up and down over a range of ±3% in 0.6% steps, and fitting gradients to the corresponding string tension and tuning length adjustment values.

For both types of Young’s modulus test, the interval between successive tension or frequency modulation steps was set to 1 min, with a total modulation cycle time of around 30 min. The results from these Young’s modulus tests are not covered in this paper since the Young’s modulus behaviour of nylon instrument strings has already been covered quite extensively in [6], published in 2017. By the time the later tests were run, there was no need for further Young’s modulus measurements, and it was more important to explore the string creep behaviour without any risk of possible interference from the tension variations applied during the Young’s modulus tests.

### 3.3. Data Processing

#### 3.3.1. Filtering to Remove Young’s Modulus Measurements

The Young’s modulus tests conducted by modulating the string tension, at the end of each applied stress step, only ran for a few hours and the corresponding data points were simply excluded prior to examining the string creep behaviour. For string N20, because the Young’s modulus was measured throughout each step by varying the target frequency, a different approach was needed. In this case, only the data points recorded at the base frequency were retained, with the time interval between the retained points averaging about 17.5 min. In the data files submitted with this paper, this pre-filtering of the response data has already been performed.

#### 3.3.2. Length Adjustment Correction Offsets

Each time a new string was mounted on one of the test rigs, and tuned up to the first target frequency, it proved impractical to measure exactly when the string first came under tension. Indeed, a certain amount of tension had to be applied before the string could be plucked effectively and its fundamental frequency determined. A tuning length adjustment offset was therefore required to correct for the length of string wound onto the winding shaft prior to the first recorded data point. The best way that had been found to estimate the required length adjustment correction offset was to fit a line to the tension and length adjustment values recorded for the very first tuning adjustment, and extrapolate the fitted gradient back to zero tension [9].

It was expected that this approach would still fail to account for some initial string creep, resulting in the estimated correction offsets being smaller than they should be. It was also suspected that this discrepancy would be larger in those cases where the first target frequency was relatively high (all strings except for N7 and N28). As will be seen later, despite these attempts to correct for the initial take-up of the string onto the winding shaft, the strain responses still appeared to show quite a range of offsets, and it was generally found to be more useful to align the strain levels at a later point in the applied stress sequences.

Due to the exponential function used to convert the length adjustment measurements into strain values (Equation (Equation 2) below), any remaining offset error in the length adjustment values will have resulted in a strain error which varied smoothly and monotonically with the strain level, rather than just a simple strain offset. Comparing the length adjustment results from the different tests, the offset errors were almost certainly no more than 4 cm (approximately one full turn of the winding shaft), and the resultant stretching of the strain values will have amounted to no more than an additional strain variation of about 1.4% across the full range of strains observed, with local deviations around each stress step response being considerably smaller. It was therefore not expected that the remaining length adjustment offset errors would have any significant effect on the analysis and interpretation of the observed creep behaviour.

#### 3.3.3. Thermal Compensation

While analysing the test results from string N7, Shiangoli noticed a correlation between the creep rate and the temperature [10]. Plotting the changes in length adjustment against the corresponding changes in temperature, during the periods of relatively steady creep towards the end of each test step, showed a particularly strong correlation. Figure 2a shows the case for the last seven days of the first stress step applied to string N7. Shiangoli identified that the underlying creep rate should not be significantly affected by these temperature changes.

Hence, the additional variations in length adjustment, required to keep the string tuned to the target frequency, must have been due almost entirely to the changes in string tension caused by the combination of the thermal contraction of the string, due to it having a negative longitudinal coefficient of linear thermal expansion, and temperature-induced changes in the tensile Young’s modulus [6]. The gradient of a line fitted to this plot could therefore be used to provide temperature compensation estimates for the measured length adjustments. It is perhaps worth noting that the *y*-axis intercept in Figure 2a corresponds to the underlying creep rate in terms of the required length adjustment over each recording interval (mm per hour for string N7).

Figure 2b shows the fitted gradient values, plotted against the applied stress, for each of the stress steps applied to string N7, except for step 9, which did not give a good result due to a somewhat lower range of temperature deviations than for the other steps. It was not possible to repeat this form of analysis with the other strings tested in this study since the introduction of the automated temperature management facility (Section 3.2 above) for these later tests meant that the temperature deviations from the target temperature of 20 °C were not as large as for string N7. Comparable values could, however, be obtained from the test results with string N8 used in the nylon string study reported in [6], by fitting lines to plots of length adjustment versus temperature during the later parts of the constant-frequency creep-settling tests, when the string appeared to have settled, after each increase in the target frequency. These data are also included in Figure 2b. String N8 was chosen as it had the same nominal diameter as the strings used in this study, while comparable results from other strings showed that the temperature sensitivity gradients varied with the string diameter.

The two sets of data spanned a similar range of values, and both appeared to be independent of the applied stress. The dashed black line in Figure 2b is the average for the N8 data (−0.022 mm/°C). This appeared to be a fairly representative value for the thermal variation in the length adjustments, and was used to estimate length adjustment corrections to compensate for deviations from the target temperature of 20 °C.

#### 3.3.4. Strain Conversion

Since the string frequency, and hence the applied tensile stress, was maintained by winding lengths of string on and off the winding shaft, the applied strain could not simply be calculated as the ratio of the tuning length adjustments to the total string length. Instead, the string length adjustments (with the initial take-up and temperature compensation corrections applied) were converted to strain values ϵ using: (2)$$\epsilon \approx e^{x/L}-1$$
where *x* is the total of the applied tuning length adjustments, including the length adjustment correction offset described above, and *L* is the whole length of the string between the winding shaft and the clamping point at the load cell. The derivation of this equation is given in [6]. When x≪L, this equation reverts to ϵ≈x/L, but, as will be seen below, the strains encountered in this study ranged up to nearly 20%, so this small-value approximation could not be used.

### 3.4. Response Modelling

The first stage in attempting to analyse the observed creep behaviour was to try fitting models to the individual stress step responses. It was expected that it should be possible to model the stretching behaviour of the strings as an elastic spring, with Young’s modulus $E_{el}$, in series with one or more Kelvin–Voigt stages, each consisting of an elastic spring of modulus $E_{i}$ in parallel with a dashpot of viscosity $\eta_{i}$, possibly followed by a final dashpot with viscosity $\eta_{f}$ [13,14]. The spring would model the initial elastic stretching, while the Kelvin–Voigt stages would model exponentially diminishing viscoelastic stretching, normally associated with primary creep [1]. The final dashpot was included to model any remaining approximately steady rise or fall towards the end of each step. While this might be associated with secondary creep, it could also be associated with the early part of another slower viscoelastic mode (see below). Figure 3 depicts such a model with two Kelvin–Voigt stages. The need perhaps to use multiple Kelvin–Voigt stages for modelling real material samples should not be unexpected [1,14]. For the three final-year student projects which attempted to fit models to the creep responses from these tests [10,11,12], a single Kelvin–Voigt stage was used quite successfully. However, subsequent trials showed that for many steps a better fit could be obtained using two Kelvin–Voigt stages, while adding a third stage did not appear to add any benefit and simply risked compromising the subsequent analysis of the fitted coefficients.

For a linear system with *k* Kelvin–Voigt stages, in which the Young’s modulus and viscosity values remain constant, at least for the duration of the applied stress step, the additional strain $\Delta \varepsilon_{s}\left(t\right)$ exhibited by such a model, in response to a step increase in stress $\Delta \sigma_{s}$ applied at time $t_{s}$ would be: (3)$$\Delta \varepsilon_{s}\left(t\right)=\Delta \sigma_{s}\left[\frac{1}{E_{el}}+\sum_{i=1}^{k}\frac{1}{E_{i}}\left(1-e^{-\left(t-t_{s}\right)/\tau_{i}}\right)+\frac{t-t_{s}}{\eta_{f}}\right]H\left(t-t_{s}\right)$$
where $\tau_{i}=\eta_{i}/E_{i}$ is the creep or retardation time, and $H\left(.\right)$ is the Heaviside or unit step function [13]. For each Kelvin–Voigt stage, when $\left(t-t_{s}\right)$ is small the behaviour is dominated by the dashpot and the strain can be approximated as $\Delta \sigma_{s}.\left(t-t_{s}\right)/\eta_{i}$. Conversely, after enough time has passed, the behaviour will be dominated by the spring component and the strain will approach $\Delta \sigma_{s}/E_{i}$. The corresponding function fitted to each step *s* was: (4)$$\Delta \varepsilon_{s}\left(t\right)=\left[A_{0}+\sum_{i=1}^{k}A_{i}\left(1-e^{-X_{i}\left(t-t_{s}\right)}\right)+B\left(t-t_{s}\right)\right]H\left(t-t_{s}\right)$$
with $A_{0},A_{i},X_{i},$ and *B* being the fitted coefficients. The fitting was performed using the SciPy curve_fit function within a Python script [19,20].

#### Modelling Multiple Stress Steps

If the model parameters remain constant as the applied stress is varied, then the strain response to a sequence of applied stress steps could be modelled as just the sum of the responses to each step. Some cases, however, required the ability to vary the model parameters from one step to the next, for which a different approach was needed. Starting from Equation (Equation 3), the remaining strain response of each Kelvin–Voigt stage to the stress step $\Delta \sigma_{s}$ applied at time $t_{s}$, for $t\geq t_{s+1}>t_{s}$, would be: (5)$$\begin{array}{cc} \Delta \varepsilon_{i,s}\left(t\geq t_{s+1}>t_{s}\right) & =\frac{\Delta \sigma_{s}}{E_{i}}\left(1-e^{-\left(t-t_{s}\right)/\tau_{i}}\right)\\ & =\Delta \varepsilon_{i,s}\left(t_{s+1}\right)+\left[\frac{\Delta \sigma_{s}}{E_{i}}-\Delta \varepsilon_{i,s}\left(t_{s+1}\right)\right]\left(1-e^{-\left(t-t_{s+1}\right)/\tau_{i}}\right) \end{array}$$
and the response to a sequence of steps could then be calculated as: (6)$$\varepsilon_{i}\left(t_{n}\leq t<t_{n+1}\right)=\varepsilon_{i}\left(t_{n-}\right)+\left[\frac{\sigma_{n}}{E_{i}\left(\sigma_{n}\right)}-\varepsilon_{i}\left(t_{n-}\right)\right]\left(1-e^{-\left(t-t_{n}\right)/\tau_{i}\left(\sigma_{n}\right)}\right)$$
where $t_{n}$ is the time the *n*th stress step is applied, $\sigma_{n}=\sum_{s=1}^{n}\Delta \sigma_{s}$ is the sum of the applied stress steps up to and including step *n*, and $\varepsilon_{i}\left(t_{n-}\right)$ denotes the strain immediately before the *n*th stress step is applied. This equation can be shown to be equivalent to using a time offset instead of an amplitude offset, but the form shown here supports negative stress steps in a computationally more robust manner, and permits the underlying Young’s modulus and viscosity values to be changed from step to step as necessary.

A small complication arises if it is required to commence the model evaluation at some step m>1 and values of $\varepsilon_{i}\left(t_{m-}\right)$ are not available. If it can be assumed that any remaining contribution from the preceding steps can be ignored, then the start point of the evaluation can be relocated to the start of step *m* by substituting $\varepsilon_{i}^{\prime }\left(t_{n-}\right)$, starting from $\varepsilon_{i}^{\prime }\left(t_{m-}\right)=0$, in place of $\varepsilon_{i}\left(t_{n-}\right)$, and replacing $\sigma_{n}$ with $\sigma_{n}^{\prime }=\sum_{s=m}^{n}\Delta \sigma_{s}$. Then, for n≥m,: (7)$$\begin{array}{cc} \varepsilon_{i}\left(t_{n}\leq t<t_{n+1}\right) & \approx \varepsilon_{i}\left(t_{m-}\right)+\varepsilon_{i}^{\prime }\left(t_{n}\leq t<t_{n+1}\right)\\ & =\varepsilon_{i}\left(t_{m-}\right)+\varepsilon_{i}^{\prime }\left(t_{n-}\right)+\left[\frac{\sigma_{n}^{\prime }}{E_{i}\left(\sigma_{n}\right)}-\varepsilon_{i}^{\prime }\left(t_{n-}\right)\right]\left(1-e^{-\left(t-t_{n}\right)/\tau_{i}\left(\sigma_{n}\right)}\right) \end{array}$$
with the component $\varepsilon_{i}\left(t_{m-}\right)$ in practice being incorporated into an amended offset $A_{0}^{\prime }$. The calculation for the series dashpot is more straightforward with: (8)$$\varepsilon_{f}\left(t_{n}\leq t<t_{n+1}\right)=\varepsilon_{f}\left(t_{n}\right)+\frac{\sigma_{n}}{\eta_{f}\left(\sigma_{n}\right)}\left(t-t_{n}\right)$$
and relocating the start time of the evaluation is much simpler: (9)$$\begin{array}{cc} \varepsilon_{f}\left(t_{n}\leq t<t_{n+1}\right) & =\varepsilon_{f}\left(t_{m}\right)+\varepsilon_{f}^{\prime }\left(t_{n}\leq t<t_{n+1}\right)\\ & =\varepsilon_{f}\left(t_{m}\right)+\varepsilon_{f}^{\prime }\left(t_{n}\right)+\frac{\sigma_{n}}{\eta_{f}\left(\sigma_{n}\right)}\left(t-t_{n}\right) \end{array}$$
with $\varepsilon_{f}^{\prime }\left(t_{m}\right)=0$, and the component $\varepsilon_{f}\left(t_{m}\right)$ also incorporated into the amended offset $A_{0}^{\prime }$.

## 4. Results

### 4.1. Observed Behaviour

Figure 4a shows the first eight steps of the stress sequences applied to strings N21 and N27, while Figure 4b shows the resulting strain responses. Three line types have been used: solid lines denote steps where the applied stress exceeded the previous maximum for that string; dashed lines denote steps where the applied stress was the same as the previous maximum; and dotted lines denote steps where the applied stress was below the previous maximum.

Looking first at string N21 (blue lines), following the initial step at close to 200 MPa, the stress was stepped down by two steps of about 30 MPa each, and then stepped back up to the original level. The total duration of these first five steps was 42 days. The stress was then increased, beyond its previous maximum of 200 MPa, by a further 30 MPa, before being stepped down by two steps. The stress sequence applied to string N27 was very similar, using very nearly the same applied stress levels, except that the increase in the applied stress beyond the previous maximum level came much earlier at step 2, after just 11 days.

Looking at the strain responses for these two strings, the key features described earlier in the Introduction can clearly be seen. When the applied stress was below the previous maximum, the strain responses to the different stress steps were all very similar, in terms of their magnitude and timing, for both positive and negative stress steps, and for both strings. The form of these responses strongly suggests that the string creep response to stepped stress changes below the previous maximum applied stress was both fully reversible, and similar across different strings.

In contrast, when the applied stress was increased beyond the previous maximum, both strings displayed a much larger strain response, apparently indicating episodes of non-recoverable plastic stretching. The big difference between the two string responses was the magnitude of the step 2 response of string N27, compared to that for string N21 step 6. In both cases, the applied stress was increased to about the same level at about 230 MPa, but the strain response of string N27 was around twice that of string N21. The only clear difference was that the stress increase from around 200 to 230 MPa was applied after 42 days and following periods of lower applied stress for string N21, but after only 11 days with no intervening stress reduction for string N27. It appeared that allowing string N21 time to settle had somehow constrained the plastic creep response when the applied stress was next increased beyond its previous maximum.

Moving beyond just these two string responses, Figure 5a,b show the full set, and full durations, of the applied stress sequences and the corresponding strain responses. To aid in comparing the results, the string responses shown in Figure 4b and Figure 5b have had vertical (strain) alignment offsets added. The common alignment point chosen was the strain reached at the end of the first step with an applied stress of close to 200 MPa (corresponding to a target fundamental frequency of 425 or 429 Hz). For most of the strings, this was the strain reached at the end of the first step, while for strings N7 and N28, it was the strain reached at the end of steps 7 and 11, respectively. The reference level was taken as the strain at the end of string N7 step 7. The negative strain values shown in Figure 5b during the first step response for string N28 were simply a consequence of this alignment process (Figure 6 provides a comparison of the unaligned strain responses).

Prior to making this alignment, the strain responses for the different strings appeared to show a significant range of initial offsets, despite the attempts to estimate the length adjustment correction offsets required to account for the initial take-up of the strings onto the winding shafts (Section 3.3.2), and even though the tests all started with the same step 1 target frequency of 429 Hz, except for strings N7 and N28, which started at 174 Hz. These offsets may have been due in part to residual errors in estimating the length adjustment correction offsets, but may also correspond to different degrees of initial straightening and settling within the strings (this will be discussed further in Section 4.2.3). As discussed in Section 3.3.2, the range of these offset variations was not large enough to have a significant effect on the response analysis. The thermal compensation adjustments were included for string N21 step 10, despite the temperature being deliberately increased during this step, since these adjustments were intended to compensate for the non-creep-related changes in the string stretching.

The tests run with strings N20, N21, and N27 were designed primarily to investigate what happened when the applied stress was stepped up and down at levels at or below the previous maximum, resulting in the episodes of fully recoverable creep already described. The responses of different strings to similar sequences of stress changes showed a high degree of consistency. The eight steps applied to string N20 had the same target frequencies, and hence very nearly the same applied stress levels, as for the first eight steps applied to string N27. Their responses can be seen to be very similar except for a steady fall in the response of string N20 (discussed below in Section 4.2.2). The later parts of the strain responses for strings N21 and N27, beyond the first eight steps shown in Figure 4b, continued to display very consistent, and fully recoverable, creep behaviour in response to applied stress steps at stress levels below the previous maximum. For string N27, this behaviour remained consistent even when the applied stress was lowered by a considerable extent, as far down as around 100 MPa.

The effect of increasing the temperature at step 10 (day 76) for string N21 can also be seen. The increased temperature clearly triggered an additional creep episode, but otherwise appeared to have no noticeable effect on the subsequent strain behaviour of the string. Figure 5c provides an expanded view of the strain response for that step together with a scaled plot of the temperature. The dip in the temperature response towards the end of day 76 corresponds to the test rig control program halting due to adjustment hunting around the required tuning point.

It is interesting that the additional creep in response to a relatively small increase in temperature at day 78 is much larger than the creep in response to the initial, much larger, increase in temperature at day 76. This presumably indicates some form of threshold effect that had to be overcome when the temperature was first increased. Another feature worth noting is that the temperature-induced creep did not cease as soon as the temperature was lowered back to its normal level, even allowing for thermal lags. The strain started increasing only a short time, much less than a day, after the temperature was increased, but had not fully levelled out by the end of the step, approximately three days after the temperature had been reduced. Whatever creep was triggered by the increase in temperature ran to its conclusion even after the temperature stimulus had been removed.

The test with string N7, the first of the set, was primarily designed to explore the string behaviour, and in particular its Young’s modulus, over a wide range of stress levels. The tests with strings N26, N33, and N34 were designed to explore the factors affecting the extent of the plastic creep. They started with the same two applied stress levels as for string N21, but varied the extent to which the applied stress was subsequently stepped down, and the sequence timing. The test with string N28 aimed to replicate the overall applied stress range used with string N7, while also exploring the factors affecting the extent of the plastic creep.

It was stated in the Introduction that each time a string was subjected to an applied stress or temperature greater than the historic maximum, the string would start creeping again and would typically take weeks to settle. This is graphically illustrated in Figure 5b by the first step response of string N34, which was still creeping after six weeks.

Since the applied stress sequence for string N7 consisted of a series of ever-increasing steps, with each step lasting for 6–8 days, it provided a form of baseline reference for the investigation of the plastic creep behaviour. The dotted red horizontal lines in Figure 5b delimit the extent of the strain response of string N7 step 8 (the step following the string N7 step with a target frequency of 425 Hz). The lower dotted red line marks the vertical alignment level described above.

It is useful to look at the behaviour of the other tested strings in this marked range. The amplitude of the step 2 strain response for string N20 was very similar to that for string N7 in response to essentially the same change in the applied stress, and indeed the same was true for the subsequently tested strings N27 and N33, which had the same first two step target frequencies as string N20. The only obvious difference was that the response for N27 appeared to be a little slower than the others. As remarked earlier, the strain response for string N21 step 6 was very much less than for these other four strings, in response to the same increase in the applied stress. Reapplying the same stress level to string N21 during steps 11 and 15 still did not produce anything like the same degree of stretching, even with the additional strain from the extra heating at step 10. Discounting the additional strain from step 10, the strain by the end of steps 11 and 15 was only about half what might have been expected.

Looking at the responses for string N26 and string N28 steps 3–6, reducing the stress down by one step, and thereby lengthening the time before the applied stress next exceeded its previous maximum (from 8 days to 21 days for string N28 steps 3–6, and from 14 to 30 days for string N26) did not appear to significantly constrain the subsequent plastic creep.

However, from the results with string N33 and string N28 steps 12–16, as well as string N21, it appeared that decreasing the applied stress by two steps, and extending the time from 8–14 days to somewhere in the range of 39–42 days, did constrain the extent of the plastic creep when the stress was next increased beyond its previous maximum. There was even some indication, from string N28 step 17, that the effects of this constraint might apply to more than one subsequent stress increase, although the model fitting results to be presented later in Section 4.2.3 suggested that this was not in fact the case.

The results from string N28 steps 7–10, decreasing the applied stress by a double step and extending the time from 7 to 29 days, also appeared to show a possible reduction in the step 10 strain response, but this was less clear. It will be shown below, also in Section 4.2.3, that the apparently reduced step 10 response may in fact have been due simply to intrinsic differences in the stretching responses of strings N7 and N28.

The results from string N34 showed that the plastic creep constraint could also be induced just by waiting for a sufficient time (42 days again) before the further increase in the applied stress. Whether or not reducing the applied stress helped to induce the subsequent plastic creep constraint was not clear, but it certainly was not necessary, and did not, on its own, appear to be sufficient. The deciding factor appeared to be the length of time before the applied stress was further increased.

These patterns of behaviour would appear to indicate rather clearly that some form of additional bonding or restructuring was slowly taking place in the string after the main part of the strain increase associated with plastic creep had ceased. Furthermore, this ‘strengthening’ process did not appear to be curtailed, and may even have been enhanced, by stepping the applied stress back down and up again.

When the previous maximum stress level was reapplied, after a period of stress reduction, it appeared that in many cases the plastic stretching continued. This was most noticeable for string N33 step 6, string N27 steps 6 and 15, and string N28 steps 9 and 15. Whatever the plastic creep constraint mechanism was, it did not appear to affect the plastic creep triggered by the preceding increase in the applied stress level, only that associated with the next increase in the maximum applied stress.

Figure 6 shows the strain at the end of each stress step plotted against the applied stress for that step. The same line styles (solid, dashed, dotted) have been used as for Figure 4 and Figure 5, but the strain offsets used to align the plots in Figure 4b and Figure 5b have not been applied. The dot-dashed black line shows the comparable data for string N8, used in the nylon string study reported in [6]. This plot for string N8 is not quite the same as the version shown in [6], because the length adjustment correction offset for that string has been recalculated using the method described in Section 3.3.2, and the plot extended to include an additional datum at the point where the string broke. It would be more conventional to plot stress against strain but in this context, with the strain being the measured response to the applied stress, it was more appropriate to plot strain against stress.

The strain values for string N8 were recorded when the string appeared to have stopped creeping. For the strings in this present study, however, the strings had usually not finished creeping by the end of each stress step. It would be tempting to think that the strain versus stress plots for these strings should therefore always lie below that for string N8, but this was not generally the case, except for strings N20 and N21. While it is very likely that the length adjustment correction offsets were not entirely accurate, any remaining offset errors, including for string N8, would not explain the range of strain–stress responses shown in Figure 6. It seems much more likely that the overall stretching behaviour of the different strings was simply different, despite these strings being made in the same way from the same material and having the same nominal diameter. It is interesting to note, though, that the strings that were tested to their breaking points (marked by thick crosses), including string N8, all reached fairly similar stress and strain levels, despite their very different stretching histories.

A striking feature of Figure 6 is the linearity, and the similarity of the gradients, of those parts of the responses where the applied stress was stepped down and back up at levels not exceeding the previous maximum applied stress (dashed and dotted line segments). This further indicates that the creep during these steps was fully reversible, and that this aspect of the string behaviour was very similar across the different strings. The gradients during the first two reduced-tension episodes with string N28 (steps 3–5 and 7–9) were slightly steeper than for the other tension-reduction episodes across the various strings. The variant behaviour of string N28 during these episodes will be explored further in Section 4.2.2 and Section 4.2.3. The distinct difference between the gradients over the reduced-tension sections and the gradients for the steps where the applied stress did exceed the previous maximum (solid line segments) gives a very clear visualisation of the different creep behaviour in these two situations.

### 4.2. Modelling and Analysis

This section describes the results obtained from fitting a succession of mathematical models of increasing complexity to the measured strain responses. The first stage was to fit a simple model to each individual stress step response, and hence estimate the elastic stretching component of the strain responses (Section 4.2.1). A multi-step model was then applied to sections of the strain responses which appeared to display the recoverable creep behaviour (Section 4.2.2). Finally, and more speculatively, the multi-step model was extended to explore the plastic creep behaviour (Section 4.2.3).

The purpose of fitting models is to gain insights into the mechanisms underlying the observed behaviour. As the complexity of the models grows, however, the number of parameters to be explored goes up, and this brings two problems. First, the process of best-fitting these parameters becomes increasingly uncertain, with multiple local optima. This makes it much more difficult to find and recognise the global best fit. The second problem is that increasing doubt is cast on whether the chosen model, even if it is the global best fit, is a true representation of what is actually happening. Other local optima with significantly different parameter values may give fits that are almost as good.

#### 4.2.1. Elastic Stretching

The model described in Section 3.4 and Equation (Equation 4) was fitted to the strain versus time response for each individual step of the stepped-stress tests. Two Kelvin–Voigt stages were used in those cases where adding a second Kelvin–Voigt stage significantly improved the fit; otherwise, just one Kelvin–Voigt stage was used. The corresponding elastic stretching component $A_{el}$ for each step was then estimated as the difference between the fitted $A_{0}$ coefficient for that step and the strain immediately before the corresponding stress step was applied. These individual step estimates could then be summed to give estimates of the total elastic strain.

For strings N7 and N21, a consequence of the tension modulation cycles applied at the end of each step to measure the Young’s modulus (Section 3.2) was that the string frequency at the end of the Young’s modulus test was usually slightly different from the target frequency for that step. This in turn meant that the strain level recorded at the start of the next step, immediately before the next applied stress change, was slightly wrong. To attempt to correct for this, the fitted model for each step was used to estimate the expected strain level at the start time of the next step, prior to calculating the elastic stretching components.

Similar corrections were not required for string N20, since the frequency modulation used with that string, to continuously monitor the Young’s modulus, always finished back at the required target frequency, while the tests run with the other strings did not include any form of Young’s modulus test and ran directly from one target frequency to the next, as listed in Table 1.

For string N21 step 10, the form of the strain response to the period of elevated temperature was different to the other steps (Figure 5c), so in this case the expected strain level, at the start of step 11, was estimated by fitting a slightly different function, equivalent to a single Kelvin–Voigt stage, to just the tail section of the step 10 response.

Figure 7 shows the total elastic strain $\sum_{s=1}^{n}A_{el}$ plotted against the applied stress $\sigma_{n}$. As for Figure 6, this is again the opposite of the normal convention, but logical in this context. To aid in the comparison, the plots have been aligned vertically in the same way as for Figure 4b and Figure 5b, with the common alignment point being the strain reached at the end of the first step with an applied stress of close to 200 MPa (corresponding to a target fundamental frequency of 425 or 429 Hz), and the strain at the end of string N7 step 7 used as the reference level. The remaining spread in the responses at around 200 MPa is due to the applied stress sequences for most of the strings passing through 200 MPa on multiple occasions (Figure 5a), each time with slightly different strain response levels.

The responses clearly displayed a common trend, with the elastic stiffness increasing with the applied stress and strain, as indicated by the gradual flattening of the strain versus stress responses.

From the tensile Young’s modulus tests conducted at the end of each step with string N7 (Section 3.2), it became clear very early on that these nylon instrument strings experienced significant strain hardening, with the Young’s modulus increasing with the applied stress. Subsequent studies [6] led to an expression for the measure of Young’s modulus $E_{T}$ at 20 °C, obtained from tension-modulation tests: (10)$$E_{T}\approx 32.0+0.0353\;\sigma_{N}-0.0269\;\rho_{N}\;\mathrm{GPa}$$
where $\sigma_{N}=4\rho_{N}L_{V}^{2}f_{1}^{2}$ (expressed in MPa) is the notional stress, and $\rho_{N}$ (expressed in kg/m^3^) is the bulk density of the unstretched string.

Murphy [11] found a reasonable match between the fitted stiffness of the series spring in some of his models and the corresponding value of $E_{T}$. If, as might be expected, the Young’s modulus for the elastic stretching obeyed a similar relation $E_{el}\left(\sigma_{N}\right)=K_{el}+G_{el}\;\sigma_{N}$, then the total elastic strain up to an applied stress σ should be: (11)$$\varepsilon_{el}\left(\sigma \right)=\int_{0}^{\sigma }\frac{1}{E_{el}\left(x\right)}dx=\frac{1}{G_{el}}\ln\left(1+\frac{G_{el}}{K_{el}}\;\sigma \right).$$

Since the tests run with strings N7 and N28 spanned a wide range of stress levels, functions of this form could be fitted to their total elastic strain versus applied stress responses. Including an additional residual offset $A_{r}$ to allow for the vagaries of the initial string stretching was found to improve the fits obtained, and the last data point, where the string broke, was excluded in both cases. The dashed lines included in Figure 7 show the fitted functions. The fitted coefficients were not an exact match for the values predicted by Equation (Equation 10), but they were reasonably close, especially for string N7, and the quality of the fits ($r^{2}$ = 0.9996 for string N7 and 0.9905 for string N28) certainly supported the use of Equation (Equation 11) as a model for the elastic stretching component $\varepsilon_{el}\left(\sigma \right)$.

#### 4.2.2. Recoverable Creep

Figure 8 explores the fitted coefficients for the Kelvin–Voigt stages from the step-by-step model fitting exercise described in the previous section. Crosses and squares indicate, respectively, steps where the applied stress exceeded or matched the previous maximum; circles indicate steps where the applied stress was below the previous maximum.

Figure 8a plots the compliance $C_{i}=1/E_{i}=A_{i}/\Delta \sigma_{s}$ against the viscosity $\eta_{i}=E_{i}/X_{i}$ for all steps except the last steps where the string broke (all strings except N20, N21, and N27). The viscosity values in Figure 8a, and later in Table 2, have been expressed in units of GPa.day. While this is not a formal SI unit, it gives an immediate intuitive appreciation of the time constants involved.

There was a clear distinction between the steps where the applied stress exceeded the previous maximum (crosses) and steps where this was not the case, with the former generally having much higher compliance values and only a modest range of relatively low viscosities, while the latter almost uniformly had lower compliance values, less than 0.1/GPa, but a much wider range of viscosities. This confirms that whatever was happening during the steps exhibiting the plastic creep episodes must have been quite different from the behaviour during the recoverable creep episodes.

Attempts to analyse the contributions from the different fitted Kelvin–Voigt stages were frustrated by the degree of variation in the fitted coefficients. Figure 8b therefore takes a different approach and plots the total amplitude of all the Kelvin–Voigt stages plus the fitted elastic strain for each step, $A_{el}+\sum_{i}A_{i}$, against the change in the applied stress, but now only for those steps where the applied stress did not exceed the previous maximum, and again excluding the last steps where the string broke. Most of the points are tightly and symmetrically clustered, supporting the view that the creep behaviour during these steps was both symmetric and fully recoverable.

The fitted line in Figure 8b shows the average gradient for all the steps where the applied stress was less than the corresponding previous maximum (circles), excluding the points for string N28 shown with the filled markers. The trend was not as clear when the fitted elastic strain was excluded from the amplitude calculations. It seems entirely plausible that, working with real measured data, sampled at discrete time points, the split of fitted amplitudes between the elastic stretching component and the Kelvin–Voigt stages may have been somewhat noisy, so that the underlying trend only appeared in its clearest form when plotting the combined amplitude in this way.

The anomalous nature of the excluded points for string N28 was already apparent in the strain plot in Figure 5b and in the gradient variations in Figure 6. For that string, the strain changes during the last period when the applied stress was reduced (starting from step 12 on day 99) were very similar to those seen with the other strings. The strain changes during the other two earlier periods of reduced stress (steps 4, 5, 8, and 9), however, were disproportionately larger. These four steps are the ones indicated with filled markers in Figure 8b. The stress reduction applied to string N27 (Figure 5a) came down almost as far (N27 step 11) as for string N28 step 8 without causing the same degree of strain reduction. Indeed, as will be seen in Figure 9c, the strain response of string N27 displayed a very high degree of consistency during the long sequence of reduced-stress steps applied to it. Perhaps the behaviour of string N28 when these earlier stress reductions were applied was influenced in some significant way by the overall stretching state of the string (explored below in Section 4.2.3), but this is only conjecture.

Following on from the above findings, Figure 9 shows some results from fitting a single model to sequences of steps where the applied stress did not exceed the previous maximum. The model was essentially that given in Equations (Equation 3) and (Equation 4), with the variation in the elastic stretching component modelled using Equation (Equation 11). The results from the step-by-step model fitting and analysis initially suggested that a single Kelvin–Voigt stage might be sufficient, but it was found that for the best multi-step fit, two stages were needed, one to model the observed recoverable creep behaviour, and one, working with the series dashpot, to model a slower trend.

The first fitted step, step *m*, for these step sequences displaying recoverable creep, was necessarily partway along the full step sequence. Consequently, fitted values of $\varepsilon_{i}\left(t_{m-}\right)$ for the Kelvin–Voigt stages, for use in Equation (Equation 6), were not available for the first fitted step. Furthermore, since the plastic creep behaviour of the preceding steps was very different from the recoverable creep behaviour being modelled, estimates for $\varepsilon_{i}\left(t_{m-}\right)$ could not sensibly be made.

The start point for the model evaluation of the Kelvin–Voigt stages and the series dashpot was therefore relocated to the start of the first fitted step, using the expressions given in Equations (Equation 7) and (Equation 9). An amended offset $A_{0}^{\prime }$ was included to cover both the initial residual strain offset $A_{r}$ and the non-elastic stretching prior to the start of the fitted steps. The model coefficients $A_{0}^{\prime },K_{el},G_{el},E_{i},\eta_{i},$ and $\eta_{f}$ were treated as constants, independent of the applied stress. Examining the single step fitting results for the steps immediately before the fitted sequences suggested that any ongoing contributions from their viscoelastic (Kelvin–Voigt) components should be very small, validating the use of Equation (Equation 7).

The fits shown in Figure 9 are for the longest available sequences of steps where the applied stress did not exceed the previous maximum. For string N21, the additional strain triggered by the temporary increase in temperature at step 10 (Section 3.2) was replaced by extrapolating the model fitted to step 9, during the step-by-step modelling exercise, over the duration of step 10, and then adjusting the strain data for the subsequent steps, so that the step 11 response followed on from the extended step 9 response. This modified strain response is shown as the dotted purple line in Figure 10b. It was found that the fit quality across the transition point between the extended step 9 and step 11 could be improved by allowing the fitting algorithm to apply a small strain level ‘tweak’ to the steps after that point. The required tweak was very small at only 0.00056.

Attempts to auto-fit the elastic stretching coefficients $K_{el}$ and $G_{el}$ were unsuccessful; the fitting function appeared to have difficulty choosing between the elastic stretching component and a Kelvin–Voigt stage. These coefficients were instead varied manually, around the values obtained from the fits for strings N7 and N28 (shown as the dashed lines in Figure 7) and those given by Equation (Equation 10), for the Young’s modulus measure $E_{T}$ from [6]. Of the combinations tried, the best results were obtained when the values of $K_{el}$ and $G_{el}$ matched those given by Equation (Equation 10).

The fits shown in Figure 9 are excellent, with the dashed red lines of the modelled responses closely matching the black lines of the measured strain responses. The fit to string N27 (Figure 9c) provided particularly strong evidence for the recoverable nature of the creep associated with stress steps at or below the level of the previous maximum applied stress, covering a wide overall range of stress levels, including one case of a double step (step 12).

The dotted blue lines in Figure 9 (offset for clarity) show the contributions from the primary Kelvin–Voigt stage, while the dotted orange lines show the combined contributions from the second Kelvin–Voigt stage and the series dashpot. For the single step fits, this slower trend would have been covered just by the series dashpot in many cases. The dotted orange line in Figure 9c also gave a particularly clear illustration of the type of lag which should be expected in the response of a Kelvin–Voigt stage having a large time constant (approximately 3 weeks in this case) and correspondingly slow response.

It was not clear whether these slower components of the fitted model were related to the previous, or perhaps ongoing, plastic creep behaviour of the studied strings, or to some other component of the strain response. It was this aspect of the string behaviour, though, that led to the cautionary statement made in the Introduction that the observed ‘plastic’ deformation may not in fact have been truly permanent. The negative slope of the response for string N20 (Figure 9a) presumably indicates some form of energy release from the previous stretching during the first two stress steps applied to that string. It is not clear why this string should have behaved quite so markedly in this way, but it is perhaps worth noting that this was the string for which a frequency modulation cycle was applied throughout its testing.

Extending the model to cover any steps where the applied stress exceeded the previous maximum did not produce good results. The model did, however, appear to be able to cover those steps where the stress matched the previous maximum in most cases. One exception was string N27 step 15 (Figure 9c), and careful examination showed that the fit actually started to fail during the preceding step 14.

For string N21, Figure 9b shows a fairly good fit result spanning steps both before (steps 7–9) and after (steps 11–15) step 10, when the additional heating was applied. While somewhat closer fits could be obtained for shorter sequences without step 10, this result does support the observation that the additional temperature-induced stretching did not appear to have any great effect on the subsequent string stretching behaviour.

Table 2 shows sets of fitted coefficients from fitting the multi-step model to different step sequences. The first five rows are for the fits shown in Figure 9 plus a sequence from string N28; the next two compare the fits obtained for shorter sequences from string N21, before and after the additional heating was applied at step 10; and the last set shows how the fitted coefficients varied as the length of the fitted sequence was changed for string N27. The $E_{1}$ and $\eta_{1}$ coefficients are for the primary Kelvin–Voigt stage, represented by the dotted blue lines in Figure 9, while the $E_{2}$, $\eta_{2}$, and $\eta_{f}$ coefficients are for the second Kelvin–Voigt stage and the series dashpot, for which the combined contribution is represented by the dotted orange lines. The viscosity values have again been given in units of GPa.day, allowing the time constants $\tau_{i}$ (in days) for the Kelvin–Voigt stages to be obtained directly by dividing the viscosity values by the corresponding Young’s modulus values.

There is considerable variation between the different strings, but the overall magnitude of each coefficient is fairly consistent. Looking at the results for string N27, fitting to different step sequence lengths, it can be seen that the coefficients varied with the number of fitted steps, but appeared to settle down somewhat as the sequences became longer. Comparing the results for the two shorter step sequences from string N21, from before and after step 10, with the longer sequence spanning step 10, the coefficients for the two shorter sequences are fairly similar, further supporting the observation that the additional heating applied during step 10 did not affect the underlying string behaviour. The coefficients for the longer sequence are rather different, and the scale of the differences is perhaps rather larger than might be expected from the different sequence lengths, based on the results just discussed for string N27, but this was presumably a consequence of residual inconsistencies following the adjustments to the step 10 response.

#### 4.2.3. Plastic Creep

The models already described for fitting the elastic stretching and recoverable creep components were simple and robust, with model coefficients that were independent of the applied stress, suggesting that the fitted models were likely to be a good match for the real underlying behaviour of the strings. Even then, as discussed earlier, it was necessary to fit the coefficients for the elastic stretching model separately from those for the recoverable creep.

The plastic stretching behaviour was quickly found to be more complex. Figure 11 explores the fitted coefficients from the step-by-step model fitting exercise for strings N7 (left) and N28 (right) as a function of the applied stress. Up- and down-triangles (▲,▼) mark the coefficients for the slower (or only) and faster Kelvin–Voigt stages, respectively, while stars (★) mark the contributions from the series dashpot.

Figure 11a,b collect together the various non-elastic amplitude contributions to the strain at each step. For the series dashpot, the values shown are the amplitude contributions at the end of each step, $B\left(t_{n+1}-t_{n}\right)$. The solid lines trace selected sequences through these coefficients, to be described in some detail shortly. The dotted vertical black lines indicate where the coefficients $A_{1}$ and $A_{2}$ of two Kelvin–Voigt stages have been summed to give the corresponding point in the marked sequence. Figure 11c,d show the cumulative amplitude (filled circles) along the sequences shown in Figure 11a,b, and the total cumulative amplitude (crosses). The dashed black lines show the results of fitting functions to these points (Equations (Equation 12) and (Equation 13) below).

Figure 11e,f show the inverse of the viscosity for the series dashpot $1/\eta_{f}=B/\sigma_{n}$, giving a measure of the dashpot fluidity, or speed of response. Figure 11g,h show the time coefficients $X_{i}$ for the Kelvin–Voigt stages, and use a logarithmic *y*-axis to better accommodate the range of values. The solid lines, and the lines jumping between subplots, trace the same sequences shown in Figure 11a,b. The marked sequence points for the second point in Figure 11g and the first four points in Figure 11h were placed at the geometric mean (the mid-point on a logarithmic scale) of the corresponding $X_{1}$ and $X_{2}$ coefficients. This geometric mean is not intended as a specific prediction; it simply indicates that the sequence was expected to run somewhere in this range.

The data selection for string N7 was straightforward, since every step appeared to exhibit plastic stretching, with no periods of reduced stress in the applied stress sequence, and no apparent instances of the plastic creep constraint seen with some of the other strings. The plots for string N7 therefore show the results for all the first nine steps, with just the last step, where the string broke, excluded. For string N28, the data selection was complicated by the presence of the reduced-tension episodes. For Figure 11b, to best display the full extent of the plastic stretching, the amplitude values shown include the contributions from any subsequent steps at less than or the same applied stress level. Hence, the values shown at 86 MPa are the sums of the amplitude contributions for steps 3–5 inclusive, and similarly at 140 MPa (steps 7–9) and 194 MPa (steps 11–15). Conversely, for Figure 11f,h, to explore the timing behaviour during the main part of the plastic creep, only the values from those steps where the applied stress exceeded the previous maximum are shown; so the results from steps 4 and 5, 8 and 9, and 12–15 were excluded. The last step, where the string broke, was also excluded.

Looking first at the total cumulative amplitude plot (crosses) for string N7 in Figure 11c, there was a clear inflection point at around 100 MPa, indicating some sort of transition in the stretching behaviour. Looking at the amplitude contributions from the series dashpot (stars) in Figure 11a, there appeared to be a discontinuity, with the dashpot contribution rising for the first four steps, and then dropping back to start again from the fifth or sixth step. A corresponding discontinuity was present in Figure 11e, with the fluidity of the dashpot generally increasing (speeding up) with the applied stress, but again restarting at the discontinuity.

This behaviour suggested that the fitted dashpot might in fact be modelling the early behaviour of two different stretching components (see Section 3.4). Looking again at Figure 11a, there was a minimum in the amplitude of the $A_{1}$ coefficients (up-triangles) for the slower (or only) fitted Kelvin–Voigt stage at the same point as the discontinuity in the dashpot contributions. A similar transition could be seen in the $X_{1}$ time coefficients (Figure 11g), with these coefficients again generally increasing (speeding up) before dropping back at the discontinuity, after which there was no clear trend.

From these observations, it was hypothesised that the plastic creep consisted of three separate stretching components, indicated by the linked sequences shown in Figure 11a: an initial component modelled by the Kelvin–Voigt stages for the first four steps, and continuing on to the faster of the two Kelvin–Voigt stages at step 5; a second component initially modelled by the series dashpot for the first four steps, and then being modelled by the slower (or only) Kelvin–Voigt stage from step 5 onwards; and a third component modelled by the series dashpot from around step 6 or 7.

The behaviour for string N28 was somewhat less clear than for string N7, especially in the timing behaviour, but a similar pattern could be discerned. In particular, the same discontinuity could be seen in the dashpot contributions, in the same stress range between the fourth and fifth points, and a similar minimum could be seen in the $A_{1}$ coefficient values at the fifth point, with a smooth progression from the dashpot amplitude contributions prior to that point.

The analysis for string N28 was complicated by the presence of many more steps where two Kelvin–Voigt stages were fitted. The most likely explanation for why this should be the case was simply the difference in the retuning and sampling intervals for the two strings: hourly for string N7 and every five minutes for string N28. Indeed, the primary reason for shortening the retuning period following the testing of string N7 was a concern that the hourly sampling rate was simply not fast enough to capture the faster aspects of the stretching behaviour. The apparent constraint of the plastic creep for some steps would also have reduced some of the amplitude contributions.

Looking at the cumulative amplitude plots for the selected sequences (Figure 11c,d), it appeared that the three stretching components were similar for both strings, and could be characterised as three phases in the overall stretching history. During the first phase, which will be referred to as the initial ‘straightening’ phase, the cumulative amplitude of the fitted $A_{i}$ coefficients, and hence the underlying compliance, increased with the applied stress in a diminishing manner to reach a limit somewhere in the region of 170–240 MPa. This could be modelled as a function of the applied stress using a single Kelvin–Voigt stage of amplitude: (12)$$A_{P1}=\frac{\sigma_{n}}{E_{1}\left(\sigma_{n}\right)}=\max\left\{0,A_{\alpha }\left(1-e^{D_{\alpha }\left(S_{\alpha }-\sigma \right)}\right)\right\}$$
where $A_{\alpha },D_{\alpha },S_{\alpha }$ were all constants. This function is zero for $\sigma \leq S_{\alpha }$, rising to a limit of $A_{\alpha }$ for $\sigma >S_{\alpha }$. The time coefficients during this phase could be modelled approximately as a linearly increasing function of the applied stress.

During the second stretching phase, there was virtually no contribution during the first two steps (with an applied stress of less than 65 MPa), after which the cumulative amplitude increased slowly at first before rising almost linearly with the applied stress from step 5 or 6 onwards (above about 160 MPa), essentially taking over from the initial straightening phase and giving rise to the inflection points in the total cumulative amplitude plots (crosses) in Figure 11c,d. This could also be modelled as a function of the applied stress using a Kelvin–Voigt stage, this time of amplitude: (13)$$A_{P2}=\frac{\sigma_{n}}{E_{2}\left(\sigma_{n}\right)}=A_{\beta }\ln\left(1-e^{-D_{\beta }S_{\beta }}+e^{D_{\beta }\left(\sigma -S_{\beta }\right)}\right)$$
where $A_{\beta },D_{\beta },S_{\beta }$ were also constants. This ‘ln-exp’ function is zero when σ=0, rises slowly to $A_{\beta }\ln\left(2-e^{-D_{\beta }S_{\beta }}\right)$ when $\sigma =S_{\beta }$, and approaches a straight line with gradient $A_{\beta }D_{\beta }$ for $\sigma >S_{\beta }$. Such an effect might be associated with a more-or-less constant compliance linked with a stress threshold. A third stretching phase, which appeared to be similar in character to the second and could be modelled in the same way, then came in at around step 7 (above about 200 MPa). The behaviour of the corresponding time coefficients for both these phases was less clear, but could be modelled approximately as constant values.

In formulating this three-phase model, there were at least three possible sources of error. The selection of which data to include in the selected sequences is perhaps the most obvious, along with the functions chosen to model the resulting cumulative amplitude responses. The choice of how many Kelvin–Voigt stages to use during the initial step-by-step modelling could also have an impact: adding or removing a Kelvin–Voigt stage affected the distribution of the strain amplitude across the remaining model components, including the series dashpot element. Overall, though, the results were plausible: the same model was derived for both strings, the transition points occurred in the same way and at the same stress levels, and the cumulative amplitude responses for the selected sequences varied smoothly with the applied stress.

Referring back to Figure 5, string N28 stretched somewhat further than string N7 during the first few steps, while during the later steps, when not constrained, the responses of the two strings appeared fairly similar. From Figure 11c,d this difference would correspond with differences in the extent of the stretching during the initial straightening phase. If the above analysis is correct, and assuming the other strings behaved in a similar way, then the corresponding straightening phase for each string would have been largely completed in response to the first applied stress step at about 200 MPa, with the observed differences in the residual offsets corresponding to intrinsic string differences during this initial straightening, as postulated earlier.

Figure 12a shows the results of fitting a model based on the above analysis to the first nine steps of the string N7 strain response, while Figure 12b shows the results of fitting the same model to all 10 steps. The form of the model was again that given in Equations (Equation 3) and (Equation 4), with the variation in the elastic stretching component modelled using Equations (Equation 10) and (Equation 11). This time, however, since the model fitting started from the first step, the expression given in Equation (Equation 6) could be used to model the response of the Kelvin–Voigt stages to the applied sequence of stress steps, and an offset was only needed to cover the initial residual strain offset $A_{r}$.

Two Kelvin–Voigt stages were again required, each covering elements of the plastic creep across a different part of the applied stress range. This was a rather different approach to the model described earlier for the recoverable creep, in which the two Kelvin–Voigt stages were active across the full range of applied stress levels, each covering quite different aspects of the stretching behaviour (the recoverable creep and some slower stretching component).

One Kelvin–Voigt stage was used to cover the initial straightening phase, with its amplitude $\sigma_{n}/E_{1}\left(\sigma_{n}\right)$ modelled as a function of the form shown in Equation (Equation 12), and its time coefficient $1/\tau_{1}\left(\sigma_{n}\right)$ allowed to vary linearly, increasing (speeding up) with the applied stress. The second Kelvin–Voigt stage was used to cover both the second and third stretching components shown in Figure 11c,d (while not shown in those plots, the cumulative sum of these two stretching components had a very similar form to that of the second stretching component on its own). Its amplitude was a function of the form shown in Equation (Equation 13), and it was modelled as having a constant time coefficient.

It was found that good fits could be obtained without including the series dashpot. This was encouraging since experience with these types of instrument strings had shown that they always eventually stopped creeping, at least until they got close to their breaking points [6].

The solid black and dashed red lines in Figure 12 show the measured and fitted strain responses, while the dotted black and red lines show the contributions from the separate Kelvin–Voigt stages. Both fits are very good, but the fit in Figure 12a is slightly better over the later steps (steps 7–9). The contributions from the two Kelvin–Voigt stages are also different in the two plots, with the results shown in Figure 12a being a closer match to the behaviour expected from the examination of the step-by-step model coefficients (Figure 11c). It seems likely that the string stretching during the last step was accelerating towards breaking, distorting the fit over the later steps in Figure 12b.

It became clear that the fitting error surface had multiple local minima: the results obtained were sensitive to the starting point used by the fitting function. Indeed, in order to find these fits, two forms of the model had to be used initially. First, a model with fixed coefficients for the main stretching Kelvin–Voigt stage, derived from the step-by-step fitting results, was used to obtain a fit for the initial straightening phase, fitting to just the first few steps of the string N7 strain response. Then, these coefficients were fixed, and an improved fit was found for the main stretching Kelvin–Voigt stage. This process was iterated until a fit could be obtained for the full strain response, and then fine-tuned by allowing both sets of coefficients to vary.

As indicated earlier, no claim can be made regarding the physical accuracy of the amplitude expressions given in Equations (Equation 12) and (Equation 13) except that they gave a reasonably good match to the behaviour found in the step-by-step fitting results. Consequently, while the model obtained seems qualitatively plausible, it may not reflect the underlying physics correctly.

Attempts to fit a model to the measured strain response from string N28 were made more complicated by the apparent reduction in some of the step responses due to some form of constraint mechanism, and the presence of the recoverable creep episodes, which, as noted above, varied in amplitude in a more complex manner than seen with the other strings.

To explore the plastic creep constraint effect, a version of the string N28 strain response was created with the recoverable creep steps removed in a similar way to the removal of the enhanced-heating step from the string N21 response (Section 4.2.2). A model similar to that used for string N7 was fitted to this modified response, with the result shown in Figure 13a. The amplitude of the main stretching phase $A_{P2}$ was scaled by an additional creep constraint factor $0<F_{c}\leq 1$. This was initially modelled as a linear function of the time since the last overall increase in the applied stress, subject to a time threshold and constrained to the range 0–1: (14)$$F_{c}=\max\left\{0,\min\left\{1,1-A_{c}\left(t-t_{x}-t_{c}\right)\right\}\right\}$$
where $t_{x}$ was the start time of the last step with an applied stress in excess of the previous maximum, and $A_{c}$ and $t_{c}$ were constants. Each time the constraint factor was applied, the amplitudes of the subsequent steps had to be adjusted down accordingly, so that the amplitude at each step *n* was calculated as: (15)$$A_{P2,n}^{\prime }=A_{P2,n-1}^{\prime }+F_{c,n}\left(A_{P2,n}-A_{P2,n-1}\right)$$

It was discovered, however, that this constraint factor was actually only required following the last episode of reduced stress, for step 16; it appeared that the strain response during step 10, following the applied stress reduction during step 8, had not in fact been constrained. Similarly, the response during step 17 did not appear to have been constrained. The model was therefore simplified to take a single value of $F_{c}$ to use with just step 16. When the constraint mechanism was required (for step 16), it was quite significant with $F_{c}\approx 0.79$ (for the fit shown in Figure 13b), indicating fairly strongly that this form of constraint was indeed not required for the other steps.

The best fitting results were obtained when both the initial straightening and the main stretching phases were allowed to continue running during those periods where the applied stress had been reduced (modelled as if the applied stress was unchanged). This appeared to be the approach most consistent with the observed behaviour, described above in Section 4.1.

To model the full strain response for string N28, the model developed so far was supplemented by a third Kelvin–Voigt stage to model the recoverable creep episodes. This took the same approach as used earlier for the multi-step fits shown in Figure 9, using the expression of Equation (Equation 7) with $\varepsilon_{3}^{\prime }\left(t_{n-}\right)$ and $\sigma_{n}^{\prime }$ restarted from the start of each recoverable creep episode. In accordance with the earlier observations about the variable size of these recoverable creep episodes, three separate model coefficients were used for the amplitude values $1/E_{3}$, while a single common value was used for the time coefficient $1/\tau_{3}$.

Figure 13b shows the fit obtained with this combined model. The fit is not quite as precise as for some of the earlier results, but is still very good. As before, the solid black and dashed red lines show the measured and fitted strain responses, while the dotted black and red lines show the contributions from the Kelvin–Voigt stages modelling the initial ‘straightening’ and main stretching phases. The dotted blue line shows the fitted result for the recoverable creep components. As expected by this point, these had progressively falling amplitudes, with fitted coefficients $1/E_{3}$ of 0.16, 0.14, and 0.06/GPa.

The contribution from the Kelvin–Voigt stage modelling the initial straightening phase (dotted black lines) was very similar in Figure 13a,b. The contribution from the Kelvin–Voigt stage modelling the main stretching phase (dotted red lines), however, was somewhat larger in Figure 13b than in Figure 13a. Whether this was due to some interaction with the fit for the recoverable creep components, or a consequence of the way the recoverable creep components were removed from the modified strain response shown in Figure 13a, or some combination of these factors, is not known. In both Figure 13a,b, this Kelvin–Voigt stage also came in with a much stronger contribution almost from the start of the response than for that fitted for string N7. To some extent, this was expected from the analysis of the single-step fitting results, but not to the extent seen here. Why this should be the case is not known, but it may simply be a feature of the particular fitting solution obtained.

While the usefulness of the final model was reduced somewhat by the need to more-or-less tailor fit the recoverable creep components and the plastic creep constraint function, it nevertheless helped to provide further insights into these elements of the string behaviour.

## 5. Discussion and Conclusions

A number of rectified nylon harp strings, having the same nominal diameter, were subjected to different sequences of applied stress steps. Each string was tested continuously for several weeks (more than 20 weeks in one case) to allow sufficient time for the stretching responses to be clearly observed. Qualitatively, much of the observed behaviour was in accordance with established expectations. However, the quantitative data gathered here are believed to be novel, and revealed some surprises. The stretching history, including the unexpected phenomena, did not appear to make any significant difference to the breaking stress or strain of the strings. The strings that were tested to their breaking points all reached rather similar breaking stress and strain levels.

The strings displayed a combination of elastic stretching, fully recoverable viscoelastic stretching, and apparently non-recoverable plastic stretching. The elastic stretching component was the most straightforward to account for: it could be modelled very well using the expression given in Equation (Equation 10) for the variation with stress of the longitudinal Young’s modulus $E_{T}$ of nylon strings. This expression was previously reported in [6].

When the applied stress was stepped down and, later, back up without exceeding the previous maximum stress level, the strings exhibited fully recoverable viscoelastic stretching with a high degree of consistency across the different strings. This aspect of the string behaviour, following each step change in the applied stress, could be modelled rather well with a single Kelvin–Voigt stage having constant Young’s modulus and viscosity components. A step-by-step model of this form was successfully fitted to a number of stepped-stress sequences, covering a wide range of stress levels, although a second Kelvin–Voigt stage working in conjunction with a series dashpot was required to cover a slower stretching component, possibly related to the preceding, and perhaps still ongoing, plastic stretching.

Most of the strings were started with the same first stress step and so had reached very similar levels of overall strain prior to the episodes of recoverable viscoelastic stretching. However, two of the strings, N7 and N28, were tested over a wider range of applied stress levels, going through several steps before they reached the starting level of the other strings. With one of these strings (string N28), episodes of reduced stress and the associated recoverable stretching were included at earlier stages of the sequence. This revealed that the extent of the recoverable stretching was relatively larger at lower overall strain levels. This behaviour did not appear to be dependent on the absolute level of the applied stress—another of the strings, N27, was stepped down to similar stress levels without any change in its strain response sensitivity—but rather on the overall (plastic) stretching point reached by the string.

The plastic creep behaviour was more complicated, with a number of the strings displaying an unanticipated phenomenon. When the applied stress was left unchanged, or was stepped down and back up again, it was noticed that in some cases the extent of the subsequent plastic stretching, when the applied stress was next increased beyond its previous maximum, was significantly less than might have been expected. The tests revealed that this apparent plastic creep ‘constraint’ mechanism seemed to depend primarily on the length of time between successive overall rises in the applied stress, with a threshold somewhere in the range of 30–40 days. This ‘strengthening’ process did not appear to be curtailed, and may have been enhanced, by stepping the applied stress back down and up again.

The physical explanation may have its origin in the way the configurations of the polymer molecules change under stress. Polymer molecules under tensile stress tend to align parallel to the stress, with the molecules packing together more densely and forming a more ordered structure, often referred to in polymers as crystallinity. This is accompanied by an increase in the number of intermolecular hydrogen and van der Waals bonds, and a decrease in the energy of the structure. While this process is happening, the polymer is seen to creep, with the string stretching. If the string is then left to settle for a time under stress, thermal fluctuations may allow the molecules to make small adjustments to their relative positions, adding more bonds and reducing their energy still further. This means that the molecules will become more firmly locked into position, and protected against further creep.

The two strings that were tested over a wider range of applied stress levels, N7 and N28, revealed another aspect of the creep behaviour. Analysis of the coefficients fitted during a step-by-step model fitting exercise suggested that the strings displayed two distinct types or phases of plastic stretching. There appeared to be an initial phase during which the plastic stretching rose with the applied stress in a diminishing manner to reach a stretching limit (modelled by Equation (Equation 12)). This has been dubbed the ‘straightening’ phase. As the applied stress was increased, the initial straightening was overtaken by an unlimited main stretching phase which rose slowly at first before approaching a linear increase with the applied stress (modelled by Equation (Equation 13)). For the other strings tested in this study, the straightening phase would have been largely completed during the first applied stress step, so the observed plastic stretching, and the associated plastic creep constraint mechanism, would have been associated with the main stretching phase.

An initial straightening phase seems intuitively plausible, and it might be expected to vary from string to string, precluding a set of common model coefficients. The behaviour during the subsequent unlimited stretching phase could then be associated with a more-or-less constant compliance linked with a stress threshold. Multi-step models based on this analysis were successfully fitted to the full stretching responses for strings N7 and N28. However, this more complicated fitting process exhibited multiple local optima, with more than one solution that gave apparently good fits, but with different partitioning between the various ingredients of the model. The fits presented here were those that appeared most compatible with the analysis of the results from the simpler and much more robust single-step fitting exercise, but no claim is made that these accurately reflect the underlying physics.

## Figures and Tables

**Figure 1 materials-18-00223-f001:**
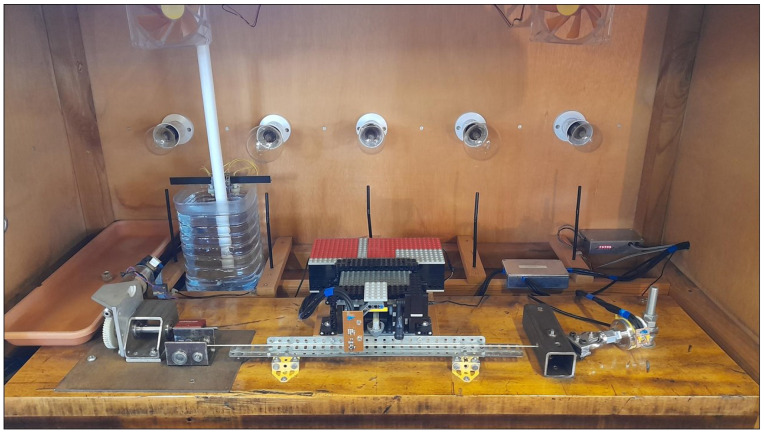
One of the two test rigs used for this study. A single string was mounted about 2 cm above a solid hardwood baseboard, with a load cell at one end to measure the string tension, and a motorised string winder at the other. The string was plucked at its mid-point, with a microphone to record the pluck responses. The temperature was controlled by switching on different combinations of incandescent light bulbs.

**Figure 2 materials-18-00223-f002:**
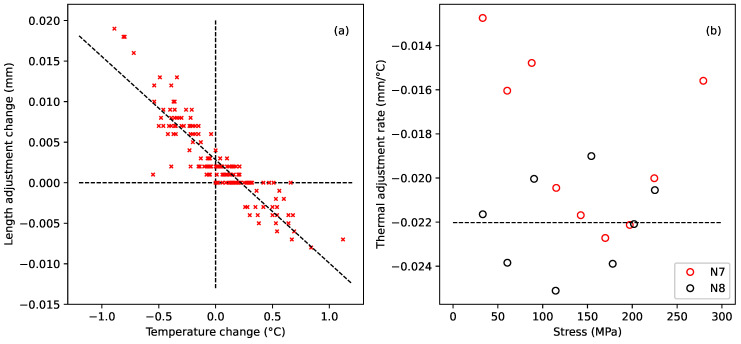
(**a**) Changes in string length adjustment versus changes in temperature for the last seven days of string N7 step 1, and (**b**) the extracted thermal sensitivity gradients versus the applied stress for the different stress steps applied to string N7, and comparable values from lines fitted to length adjustment versus temperature responses for string N8 from the previously reported study [6].

**Figure 3 materials-18-00223-f003:**
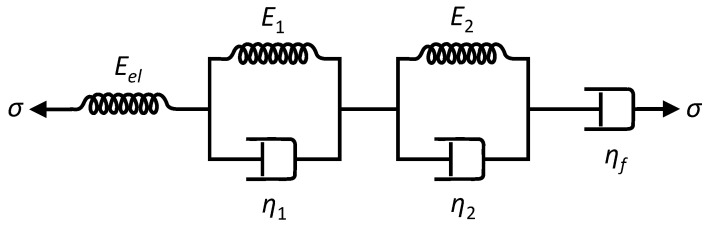
The constitutive model used in attempting to replicate the observed creep behaviour. The model consisted of a spring in series with one or more Kelvin–Voigt stages followed by a series dashpot.

**Figure 4 materials-18-00223-f004:**
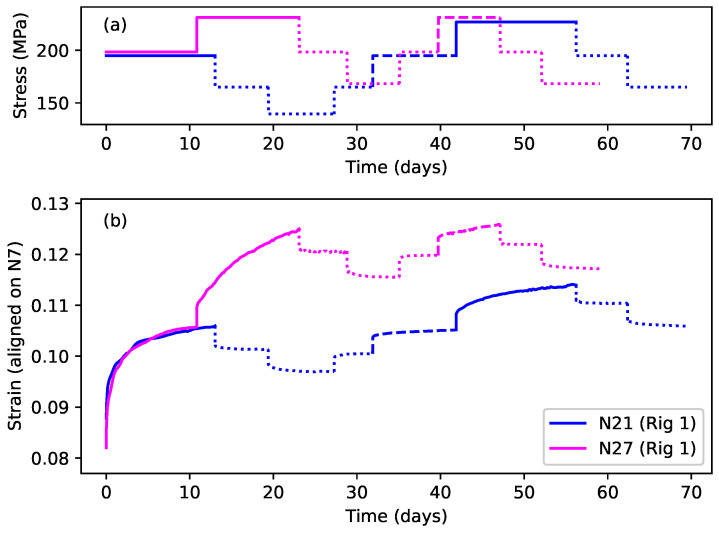
(**a**) The first eight steps of the stress sequences applied to strings N21 and N27. (**b**) The corresponding strain responses. Solid, dashed, and dotted lines denote, respectively, steps where the applied stress exceeded, matched, or was less than the previous maximum for that string.

**Figure 5 materials-18-00223-f005:**
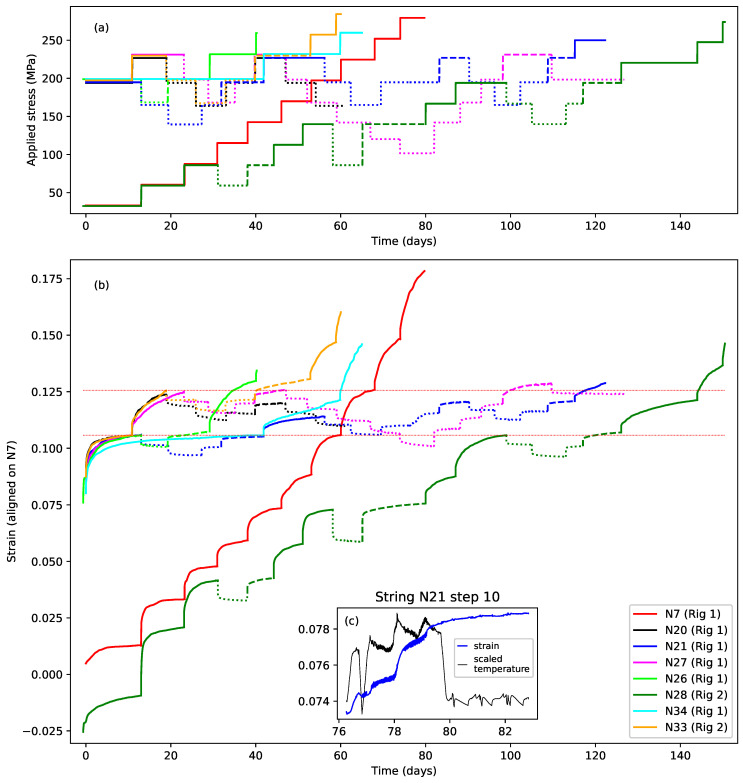
(**a**) The full set of stress sequences applied during the stepped-stress tests. (**b**) The corresponding strain responses, with alignment offsets added using the strain at the end of string N7 step 7 as the reference. Solid, dashed, and dotted lines denote, respectively, steps where the applied stress exceeded, matched, or was less than the previous maximum for that string. (**c**) The strain and (scaled) temperature responses for string N21 step 10.

**Figure 6 materials-18-00223-f006:**
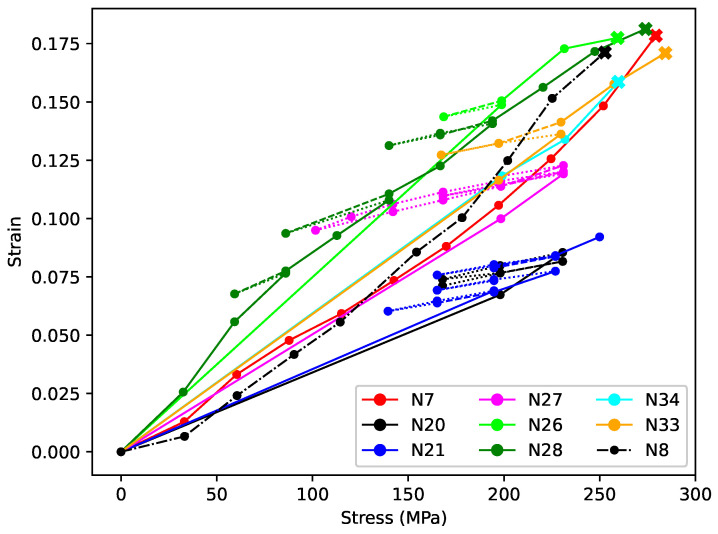
The strain at the end of each stress step plotted against the applied stress for that step (this is the opposite of the normal convention of plotting stress against strain). Solid, dashed, and dotted lines denote, respectively, steps where the applied stress exceeded, matched, or was less than the previous maximum for that string. Thick crosses mark the string breaking points. The dot-dashed black line shows the data for string N8, from the study of nylon strings reported in [6].

**Figure 7 materials-18-00223-f007:**
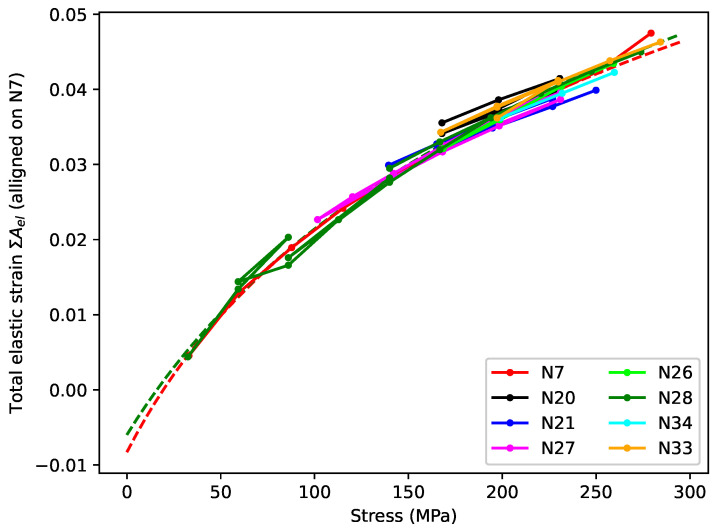
Estimates for the total elastic strain plotted against the applied stress (this is again the opposite of the normal convention of plotting stress against strain). The plots have been aligned around the 200 MPa point using string N7 as the reference.

**Figure 8 materials-18-00223-f008:**
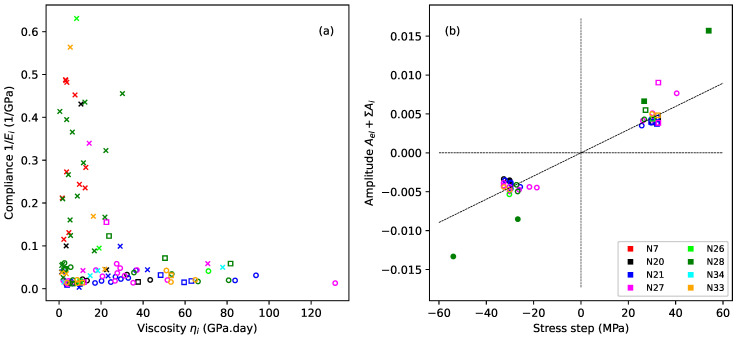
Results from the step-by-step model fitting exercise for the Kelvin–Voigt stages: (**a**) compliance versus viscosity, (**b**) the elastic strain step $A_{el}$ plus the sum of the fitted amplitudes $\Sigma A_{i}$ versus the stress step $\Delta \sigma_{s}$. Crosses and squares indicate, respectively, steps where the applied stress exceeded or matched the previous maximum; circles indicate steps where the applied stress was below the previous maximum. Subplot (**b**) only shows the results for steps below or returning to the previous maximum applied stress (circles and squares).

**Figure 9 materials-18-00223-f009:**
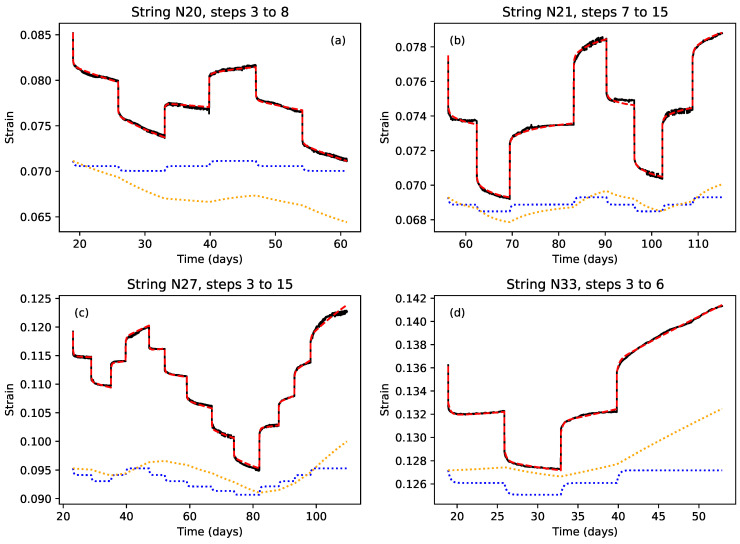
Multi-step model fitting results for step sequences where the applied stress did not exceed the previous maximum: (**a**) string N20 steps 3 to 8, (**b**) string N21 steps 7 to 15, (**c**) string N27 steps 3 to 15, and (**d**) string N33 steps 3 to 6. The black lines are the measured strain responses; the dashed red lines are the fitted model responses. The dotted blue lines show the (offset) contributions from the primary Kelvin–Voigt stage, while the dotted orange lines show the combined contributions from the second Kelvin–Voigt stage and the series dashpot.

**Figure 10 materials-18-00223-f010:**
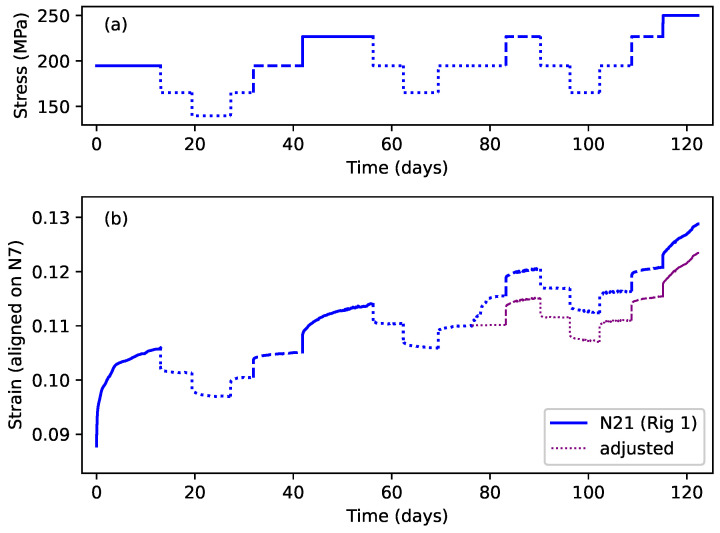
(**a**) The full stress sequence applied to string N21. (**b**) The corresponding strain response. Solid, dashed, and dotted lines denote, respectively, steps where the applied stress exceeded, matched, or was less than the previous maximum. The purple line shows the estimated effect of removing the additional temperature-induced strain from step 10.

**Figure 11 materials-18-00223-f011:**
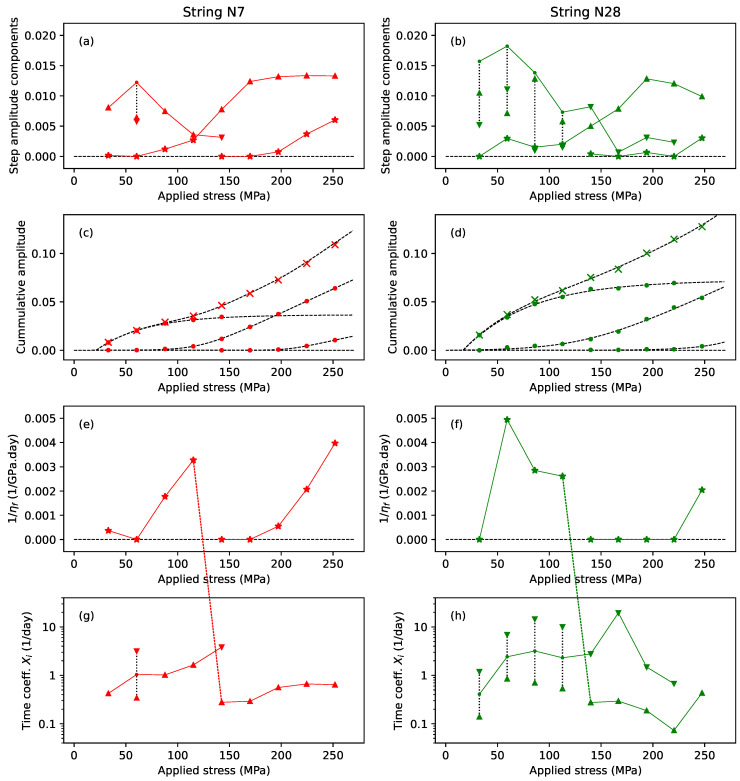
Results from the step-by-step model fitting exercise for strings N7 and N28, plotted against the applied stress: (**a**,**b**) the fitted amplitude components, (**c**,**d**) the cumulative amplitude along the linked sequences shown in subplots (**a**,**b**) and the total cumulative amplitude, (**e**,**f**) the inverse of the viscosity for the series dashpot, and (**g**,**h**) the fitted time coefficients for the Kelvin–Voigt stages. Stars (★) denote the series dashpot while up- and down-triangles (▲,▼) denote the slower (or only) and faster Kelvin–Voigt stages. Solid lines indicate the selected sequences for different stretching phases, dashed black lines are the fitted functions (see text).

**Figure 12 materials-18-00223-f012:**
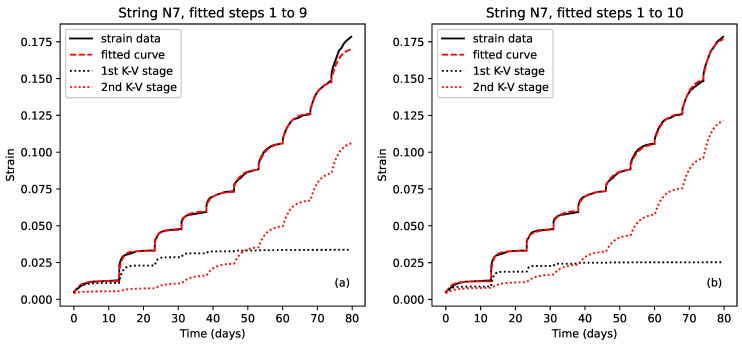
Multi-step model fitting results for string N7: (**a**) fitting to steps 1 to 9; (**b**) fitting to all 10 steps. The solid black and dashed red lines show the measured and fitted strain responses. The dotted black and red lines show the contributions from the Kelvin–Voigt stages modelling the initial ‘straightening’ phase and the main stretching phase.

**Figure 13 materials-18-00223-f013:**
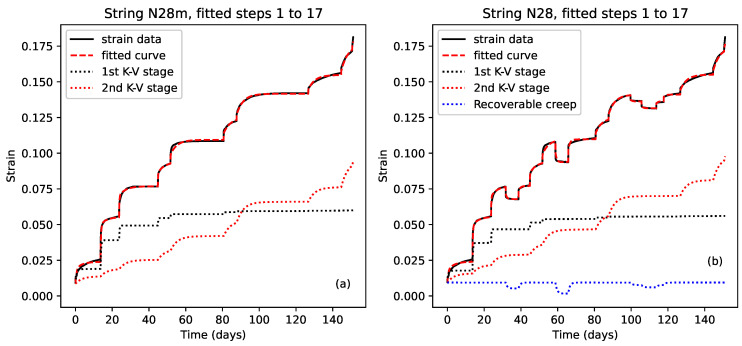
Multi-step model fitting results for string N28: (**a**) fitting to the modified strain response with the recoverable creep episodes removed; (**b**) fitting to the full measured strain response. The last (partial) step was excluded from the fit in both cases. The solid black and dashed red lines show the measured and fitted strain responses. The dotted black and red lines show the contributions from the Kelvin–Voigt stages modelling the initial ‘straightening’ phase and the main stretching phase. The dotted blue line shows the fitted result for the recoverable creep components.

**Table 1 materials-18-00223-t001:** The set of strings studied, showing the unstretched string parameters, and the target fundamental frequencies used for testing. The number in the first column provides a unique reference ID for each test string. It can be used to cross-reference with the dataset submitted with this paper, and the data archives for the previously published studies [6,7,8,9].

No.	Diameter (mm)	Density (kg/m^3^)	Test Rig	Target Test Frequencies (Hz)	Test Period
N7	0.656	1094	Rig 1 *	174, 235, 283, 324, 361, 394, 425, 453, 480, 505	Oct 2012–Jan 2013
N20 ^†^	0.660	1076	Rig 1 *	429, 463, 429, 395, 429, 463, 429, 395	Dec 2013–Feb 2014
N21 ^†^	0.645	1058	Rig 1 *	429, 395, 363, 395, 429, 463, 429, 395, 429, 429 with heating, 463, 429, 395, 429, 463, 486	Dec 2014–April 2015
N27 ^†^	0.659	1078	Rig 1	429, 463, 429, 395, 429, 463, 429, 395, 363, 334, 307, 363, 395, 429, 463, 429	February–June 2023
N26	0.656	1080	Rig 1	429, 395, 429, 463, 490	November–December 2023
N28	0.658	1080	Rig 2	174, 235, 283, 235, 283, 324, 361, 283, 361, 394, 425, 394, 361, 394, 425, 453, 480, 505	Nov 2023–April 2024
N34	0.653	1082	Rig 1	429, 463, 490	March–May 2024
N33	0.662	1078	Rig 2	429, 463, 429, 395, 429, 463, 490, 515,	April–June 2024

^†^ All the strings except for N20, N21, and N27 were tested until they broke. * The first three tests were run using the original SHT75 temperature and humidity sensors, which were replaced with the updated SHT85 sensors in late 2019 and early 2020.

**Table 2 materials-18-00223-t002:** Sets of fitted coefficients from fitting a multi-step model to different step sequences. The first five rows are for the fits shown in Figure 9 plus a sequence from string N28; the next two compare the fits obtained for shorter sequences from string N21, before and after the additional heating was applied at step 10; and the last set shows how the fitted coefficients varied as the length of the fitted sequence changed for string N27.

String	Steps	$E_{1}$	$\eta_{1}$	$E_{2}$	$\eta_{2}$	$\eta_{f}$	N21
		(GPa)	(GPa.Day)	(GPa)	(GPa.Day)	(GPa.Day)	Tweak
N20	3–8	62.6	11.9	12.2	134	−2473	
N21	7–15	76.8	19.2	32.5	149	11,090	0.00056
N27	3–15	28.1	6.8	8.0	175	1528	
N33	3–6	30.0	8.7	6.4	161	919	
N28	12–15	35.0	3.56	16.2	106	2587	
N21	7–9	46.7	10.5	15.3	191	2079	
N21	11–15	46.2	17.9	12.6	237	9674	
N27	3–5	31.7	3.0	23.9	209	4505	
N27	3–6	36.8	4.1	13.5	171	2018	
N27	3–7	41.0	3.4	14.7	154	2058	
N27	3–8	42.5	2.6	16.5	143	2319	
N27	3–9	33.8	2.9	10.5	164	1780	
N27	3–10	31.8	5.5	6.1	184	1360	
N27	3–11	31.7	5.9	5.4	186	1297	
N27	3–12	30.3	3.9	7.6	162	1448	
N27	3–13	30.4	3.5	7.8	158	1460	
N27	3–14	29.9	3.9	7.8	162	1463	

## Data Availability

Data are available in the Appendix A.

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
