# Peer review of "Plastic Creep Constraint in Nylon Instrument Strings"

_materials, 2025, doi:10.3390/ma18020223_

Round 1
Reviewer 1 Report
Comments and Suggestions for Authors
I have the following comments regarding this article:
- In the sub-section “2. Underlying physics of polymer behaviour” the authors comment that van der Waals forces are responsible for intermolecular interactions between polymer chains. However, in the case of nylon, much stronger hydrogen bonds, formed between amide units, play a significant role in the material's mechanical properties. These hydrogen forces are also responsible for nylon crystallization. Please include the hydrogen forces in the discussion.
- The dashed line is not clearly visible in Figure 4(b).
Reviewer 2 Report
Comments and Suggestions for Authors
In this manuscript, the authors investigate the creep behavior of nylon strings under different stress cycles. The authors conducted the experiment using a relatively simple setup and provided a detailed data analysis. Overall, the manuscript is interesting; however, I have some concerns and suggestions for improvement before publication.
1. In Figure 5, why does N28 show a negative strain?
2. The reviewer believes that the modeling part of this study provides a sound explanation. However, another important aspect of the validity of the analysis lies in the total recoverable creep after all the stress is removed. Do the authors have any data on the strain of the string after the test? Integrating this information would better verify the validity of the modeling.
3. The “crystallinity” explanation provided by the authors is somewhat difficult for the reviewer to accept. Why would the settlement of creep promote the growth of crystallinity? Common understanding suggests that crystallization is positively related to molecular alignment, which is typically enhanced when the chains are constantly under stress. Moreover, this explanation can be easily verified through a DSC experiment, which the reviewer suggests performing.
